# Pregnancy Outcomes in Double Stimulation versus Two Consecutive Mild Stimulations for IVF in Poor Ovarian Responders

**DOI:** 10.3390/jcm11226780

**Published:** 2022-11-16

**Authors:** Jingzhe Li, Shiqing Lyu, Shijian Lyu, Minzhi Gao

**Affiliations:** Shanghai Key Laboratory for Assisted Reproduction and Reproductive Genetics, Center for Reproductive Medicine, Renji Hospital, School of Medicine, Shanghai Jiaotong University, Shanghai 200135, China

**Keywords:** poor ovarian response, double stimulation, two consecutive mild stimulation, pregnancy outcomes, frozen embryo transfer

## Abstract

To compare pregnancy outcomes between double stimulation (DouStim) and two consecutive mild stimulations in poor ovarian responders, this study retrospectively analyzed 281 patients diagnosed as having poor ovarian response (POR) who underwent oocytes retrieval for in vitro fertilization (IVF) or intracytoplasmic sperm injection (ICSI) from January 2018 to December 2020. They were divided into two groups: the DouStim group (n = 89) and the two consecutive mild stimulations group (n = 192). The results illustrated that there were no significant differences in the number of oocytes and 2PNs between the two groups. The number of frozen embryos [1 (0, 2) versus 1(0, 2)] was significantly lower and the proportion of patients without frozen embryos (39.3% versus 26.0%) was significantly higher in the DouStim group than in the two consecutive mild stimulations group (*p* < 0.05). There were no significant differences in the clinical pregnancy rate (CPR) and the cumulative live birth rate (CLBR) between the two groups (*p* > 0.05). The intra-subgroup comparison showed that in young POR patients under 35 years old, there were no significant differences in clinical indicators and pregnancy outcomes (*p* > 0.05). In elderly POR patients aged 35 years and above, the number of frozen embryos [1 (0, 1.5) versus 1 (0.25, 2)] (*p* < 0.01) was significantly lower in the DouStim group than in the two consecutive mild stimulations group, but the pregnancy outcomes were not significantly different (*p >* 0.05). In conclusion, the DouStim protocol is inferior to the two consecutive mild stimulations protocol in terms of the number of frozen embryos, which mainly occurs in elderly patients, but there is no difference in pregnancy outcomes between the two protocols.

## 1. Introduction

Poor ovarian response (POR) refers to the reduction in ovarian reserve and the insensitivity of ovaries to gonadotropins (Gn) during ovarian stimulation therapy in infertile patients. POR has always been a challenge for in vitro fertilization and embryo transfer (IVF-ET) technology because the number of oocytes obtained after controlled ovarian stimulation (COS) will decrease. The incidence of POR in the infertile population is about 9% to 24% [1], which is characterized by decreased ovarian reserve indicators such as anti-Müllerian hormone (AMH), insensitivity to ovarian stimulation drugs, and difficulty in obtaining sufficient mature oocytes, etc. In patients with prospective POR, a high dose of Gn did not significantly improve the number of oocytes retrieved and the number of available embryos, let alone pregnancy outcomes [2]. Thus, the mild stimulation protocol with low-dose Gn and an estrogen antagonist (such as clomiphene) as adjuvants seems to be a cost-effective treatment option for POR patients [3]. In recent years, a new COS protocol called the double stimulation (DouStim) protocol has been developed and applied in POR patients. That is, in the same menstrual cycle, two consecutive ovarian stimulations and oocyte retrievals are performed in the follicular phase and the luteal phase, respectively. DouStim was first proposed by Kuang et al. in 2014, also known as the “Shanghai Protocol” [4]. Theoretically, ovarian follicles only develop during the “follicular phase”. In fact, a previous study has shown that there are two or three waves of follicular development in a menstrual cycle [5]. In addition to the biphasic stimulation regimen based on the mild stimulation protocol proposed by Kuang, some physicians have used a dual ovarian stimulation regimen based on a pituitary suppression protocol. Recruitment and collection of luteal phase follicles has been demonstrated to be feasible [6]. Unlike the traditional mild stimulation protocol where ovarian stimulation and oocyte retrieval are performed only once during the follicular phase, the DouStim protocol involves conducting it twice in one menstrual cycle. Therefore, it is inappropriate to compare clinical indicators and pregnancy outcomes between DouStim and one single mild stimulation cycle; it is more accurate to compare the DouStim protocol with the protocol of two consecutive mild stimulations. However, there is little research in this area. This retrospective study took POR patients who underwent DouStim as study objects and those who underwent two consecutive mild stimulations as controls to compare the differences in laboratory parameters and pregnancy outcomes and to explore risk factors and protective factors that may affect pregnancy outcomes.

## 2. Materials and Methods

### 2.1. Study Population and Design

This study is a retrospective analysis conducted at the Center for Reproductive Medicine, Renji Hospital, Shanghai Jiao Tong University School of Medicine (Shanghai, China). From January 2018 to December 2020, a total of 281 women with POR according to the Bologna criteria and undergoing oocyte retrieval for IVF or intracytoplasmic sperm injection (ICSI) were recruited. At that time, DouStim seemed like a novel protocol for poor responders. Therefore, some patients who approved this scheme and met specific conditions (two or more follicles ≤ 8 mm in bilateral ovaries on the trigger day in follicular stimulation) received the new treatment. Among them, 89 patients received DouStim and 192 received two consecutive mild stimulations. Patients with severe complications such as hydrosalpinx, severe endometriosis, abnormal uterine cavity morphology, untreated thyroid dysfunction, adrenal disease, hyperprolactinemia, and other endocrine diseases were excluded. Additionally, patients whose male partner had extreme oligospermia or asthenospermia (density < 1 million/mL and forward motility rate < 1%, respectively) or severe teratospermia (deformation rate ≥ 99%) were also excluded, as these conditions may affect pregnancy outcomes [7].

### 2.2. Treatment

#### 2.2.1. DouStim Protocol

Clomiphene citrate (CC, French, Gault Pharmaceuticals, Cyprus) at 50–100 mg/d ± human menopausal gonadotropin (hMG) at 75–150 IU/d (hMG or Lishenbao, Livzon Pharmaceutical, Zhuhai, Guangzhou, China) was administered to POR patients on the second or third day of menstruation for ovulation induction. When at least one follicle reached 18 mm or two follicles reached 17 mm, a gonadotropin-releasing hormone agonist (GnRH-a, Decapeptyl, Ferring Pharmaceuticals, Saint-Prex, Switzerland) at 0.2 mg ± human chorionic gonadotropin (hCG) at 2000 IU/d (Livzon Pharmaceuticals, Zhuhai, Guangzhou, China) or hCG at 6000–10,000 U was injected to trigger ovulation. Since there is no pituitary downregulation in the mild stimulation protocol, oocytes were retrieved 34 h later to avoid premature ovulation. Indomethacin (Indomethacin Suppository, Hubei Dongxin Pharmaceutical, Jinzhou, Hubei, China) was used to prevent premature follicle rupture. Based on the strategy that Kuang et al. reported in 2014 [4], luteal phase stimulation (LPS) was performed with CC at 50 mg/d + hMG at 225 IU/d on the second day after oocyte retrieval, when there were two or more follicles ≤8 mm in both ovaries on the trigger day. If the conditions were not met, another follicular phase ovarian stimulation was conducted in the next menstrual period. When the follicles reached 14 mm, progesterone tablets (Diphoton, Abbott Pharmaceuticals, Chicago, IL, USA) at 20 mg/d were added to prevent menstrual onset. The triggering protocol was the same as in the follicular phase, with oocytes retrieval 36 h later.

#### 2.2.2. Two Consecutive Mild Stimulations Protocol

Ovarian mild stimulation was performed during the follicular phase of two consecutive menstrual cycles. Ovulation-inducing drugs and triggers were the same as for follicular phase stimulation in the DouStim protocol. Indomethacin was appropriately used to prevent premature follicle rupture.

After oocytes retrieval, the number of oocytes was recorded under a microscope by a laboratory embryologist, and they were fertilized by conventional IVF or ICSI techniques. After 16–18 h, 2PN fertilized oocytes were identified as normally fertilized. The development of cleavage-stage embryos was observed on the third day after oocyte retrieval, and blastocysts were observed on the fifth and/or sixth days. Based on Gardner and Schoolcraft, high-quality cleavage-stage embryos (at least 6-cell embryos with less than 20% fragmentation) and blastocysts (developed to the fourth stage, and the inner cell mass reached grade B) were frozen. According to the management routine of our center, only one high-quality cleavage-stage embryo was frozen on the third day (D3) in all patients in each oocyte retrieval cycle, and all the remaining embryos were cultured for blastocysts.

#### 2.2.3. Frozen Embryo Transfer Protocol

As clomiphene was used in the whole process of ovarian stimulation, the development of endometria was affected. Therefore, fresh embryo transfer was not carried out and frozen whole embryos were selected. Thus, all patients’ acquired embryos were treated with frozen embryo transfer (FET), and FET was included until December 2021 in this study. Both groups were treated with hormone replacement treatment (HRT) cycles or natural cycles for endometrial preparation. Considering the convenience and reliability for patients, HRT cycles are the first choice, except for patients who have regular periods and reject HRT. In the HRT group, patients were given oral estradiol (Progynova, Bayer Pharma, Leverkusen, Germany; or Femoston, Abbott Pharma, Chicago, IL, USA) at 4–8 mg/d on the 4th day of menstruation. After 10 days of administration, endometrial thickness was monitored by B-ultrasound, and the levels of serum estrogen and progesterone were checked. When the endometrial thickness reached 7 mm, E_2_ ≥ 150 pg/mL, and P_4_ < 1.5 ng/mL, oral dydrogesterone tablets at 20 mg/d plus vaginal progesterone sustained-release gel (Crinone, Merck Group, Darmstadt, Germany) at 90 mg/d or vaginal progesterone capsules (Utrogestan, Besins, Paris, France) at 600 mg/d were administered to transform the endometrium. Cleavage-stage embryos and blastocysts were transferred on the third or fifth days after endometrial transformation, respectively. In natural cycle patients, ovulation was monitored by B-ultrasound on day 10 of menstruation. When the dominant follicle reached 18 mm in diameter and the endometrial thickness reached 7 mm, the blood estrogen, progesterone, and LH hormone levels were checked. When E_2_ ≥ 150 pg/mL and P_4_ < 1.5 ng/mL, hCG at 6000 IU was injected intramuscularly. From the second day, oral dydrogesterone tablets at 20 mg/d plus vaginal progesterone sustained-release gel at 90 mg/d or vaginal progesterone capsules at 300 mg/d were administered. Cleavage-stage embryos and blastocysts were transferred on the third or fifth day after progesterone administration, respectively. All patients maintained luteal support until 14 days after embryo transfer. Our center routinely transfers a blastocyst in the first FET and chooses two-step transfer in the second FET, which means transferal of a D3 cleavage embryo and a blastocyst, except for patients with only one type of embryo. Serum hCG level and B-ultrasound were checked to confirm a successful pregnancy. For pregnant women, the luteal support would continue until about 10 weeks of pregnancy.

### 2.3. Outcome Measurement and Statistical Analysis

The primary measurements were defined as contribution to clinical pregnancy rate (CPR) and cumulative live birth rate (CLBR). Secondary measurements were the number of oocytes, double pronuclei (2PN), and embryos obtained. In addition, we collected data including rates of early abortion, ectopic pregnancy, preterm birth, and birth defects. All pregnant patients were followed up regularly by telephone until the end of the pregnancy. The main contents of follow-up included adjustment of luteal support and other drug treatments, serum human chorionic gonadotropin (hCG) level after transplantation, B- ultrasonography, examination of fetal nuchal translucency, delivery status, termination of pregnancy, etc.

Due to the “freeze all” treatment strategy, the calculation of clinical pregnancy rate and live birth rate in one cycle was not comprehensive. We defined CPR as the proportion of pregnancies after FET per patient, and the CLBR was defined as the proportion of live births after FET per patient. POR patients who lack enough available embryos often choose to postpone embryo transfer, so this study used the number of patients who were included in all COS cycles entering ovulation stimulation therapy as the denominator [8].

All continuous variables were checked for normality. The variables that conformed to a normal distribution were described as mean ± standard deviation (mean ± SD), and the t-test was used for comparison between groups. Continuous variables with a skewed distribution were described as medians (25th quantile to 75th quantile) [M (P25, P75)], and comparison between groups was performed using the non-parametric rank-sum test. Enumeration data were expressed as rate (%), and the chi-square test was used for comparisons. Tests for multicollinearity and linearity were performed on variables considered potentially clinically significant, and logistic regression analyses were performed to explore factors influencing pregnancy outcomes in POR patients. Statistical analysis was performed using IBM SPSS 25.0 software. *p* < 0.05 was defined as statistically significant.

## 3. Results

### 3.1. Basic Characteristics and Distribution of the Study Population

A total of 281 POR patients undergoing IVF/ICSI-FET for infertility were included in this retrospective analysis. They were grouped according to different protocols of ovarian stimulation (the DouStim group and the two consecutive mild stimulations group). There were no significant differences in baseline indicators such as age, BMI, duration, types of infertility, basal endocrine level, AFC, and fertilization type between the two groups (*p* > 0.05) (see Table 1).

#### 3.1.1. Observation Indexes of Ovarian Stimulation

We collected data from two ovarian stimulation cycles for each patient. This study found that the Gn duration and dosage; the number of follicles (>14 mm); and the serum levels of LH, E_2_, and P_4_ on the trigger day in the DouStim group were significantly higher than those in the two consecutive mild stimulations group (*p* < 0.01). There was no significant difference in the number of oocytes retrieved and 2PNs between the two groups (*p* > 0.05). The DouStim group had significantly lower numbers of frozen embryos and a higher proportion of patients without frozen embryos than the two consecutive mild stimulations group (*p* < 0.05) (Table 2).

#### 3.1.2. Pregnancy Outcomes

There were no significant differences between the two groups in terms of the FET times, the cumulative number of transferred embryos, the proportion of natural cycles (NC), the proportion of blastocysts transferred, and the proportion of patients with residual embryos to be transferred. Additionally, the distribution of patients who were treated for natural cycles showed no difference between the two groups (*p* > 0.05). There were no significant differences in CPR and CLBR between the DouStim group and the two consecutive mild stimulations group (*p* > 0.05). There were also no significant differences in the rates of early abortion, ectopic pregnancy, neonatal preterm birth, and birth defects including congenital heart disease and hypospadias between the two groups (Table 3).

#### 3.1.3. Multivariate Logistic Regression Analysis of Indexes Affecting CPR or CLBR

To control the confounding factors, some relevant indexes with clinical significance were included in the multivariate logistic regression analysis. Considering the physiological characteristics of the corpus luteum, the initial dosage and duration of Gn in the luteal phase stimulation protocol were routinely higher than those in the mild stimulation protocol, and the P_4_ level on the trigger day was also significantly higher than in the follicular phase, so these indexes were not included in the regression analysis. The proportion of patients without frozen embryos was also not included in the regression equation because the number of patients was too small. The results showed that basic characteristics such as basal endocrine level, infertility duration, infertility type, and the stimulation protocol did not significantly affect pregnancy outcomes. Age may be a risk factor for clinical pregnancy rate (*p* < 0.05) and live birth rate (*p* < 0.01) in POR patients (Table 4 and Table 5).

#### 3.1.4. Comparison of Clinical Observation Indexes and Pregnancy Outcomes among Different Age Subgroups of POR

The intra-subgroup comparison showed that in young POR patients under 35 years old, there were no significant differences in the number of oocytes, 2PNs, and frozen embryos or in the number and times of embryos transferred between the two protocols, and the pregnancy outcomes, including CPR and LBR, were also not significantly different (*p* > 0.05) (Table 6). In elderly POR patients aged 35 years and above, there were no significant differences in the number of oocytes and 2PNs between the two protocols (*p* > 0.05), but the number of frozen embryos was significantly lower in the DouStim group than in the two consecutive mild stimulations group [1 (0, 1.5) versus 1 (0.25, 2)] (*p* < 0.01), and the CPR and LBR were not significantly different between the two protocols (*p* > 0.05) (Table 7).

## 4. Discussion

Our study population is not completely consistent with the POSEIDON classification, so the Bologna criteria were adopted as the diagnostic standard for POR. Previous studies have shown that POR patients who underwent DouStim in “one menstrual cycle” harvested more oocytes and usable embryos. This may improve the CLBR in patients diagnosed as “expected POR” [9]. DouStim involves two oocyte retrieval operations, so some scholars have questioned whether there is a difference between complete DouStim and two conventional ovarian stimulations. It still needs further investigation [10]. Therefore, the purpose of this study was to compare DouStim and the two consecutive mild stimulations protocol in the POR population. Due to the high incidence of aneuploid embryos in elderly patients [11], improvement of the ovulation stimulation protocol alone often has no significant effect on pregnancy outcomes. Therefore, this study did not include elderly patients over 40 years old with POR. Overall, the study population can be categorized as POSEIDON groups 3 and 4, except for patients over 40 years old. 

This study showed that in the two groups of POR patients with the same baseline indicators, the duration and total dose of Gn in the DouStim group were higher than those in the two consecutive mild stimulations group. The physiological characteristic of lower levels of endogenous gonadotropins in the luteal phase makes the LPS need a higher initial dose of Gn (225 U/d) and a longer duration [12]. The endocrine conditions (such as LH, E_2_, and P_4_) on the trigger day were also different; for example, the serum P4 level on the trigger day was significantly higher in the DouStim group, which was related to the luteal phase. In addition, while there was no difference in the number of oocytes retrieved between the two groups, the number of frozen embryos in the DouStim group was lower than that in the two consecutive mild stimulations group, and the proportion of patients without frozen embryos was higher, which means that the efficiency of the DouStim protocol in obtaining available embryos was lower than that of the two consecutive mild stimulations protocol. In 2014, Kuang et al. reported for the first time that 38 POR patients had undergone dual ovarian stimulation. The number of high-quality embryos and frozen embryos and the cleavage rate of fertilized oocytes obtained showed no difference between the double phases. However, the number of oocytes and usable embryos obtained in LPS was more than that in FPS. Therefore, they believed that DouStim could improve pregnancy outcomes [4]. Vaiarell and Jin argued that POR patients undergoing the DouStim regimen may improve their pregnancy outcomes because of the relatively high oocyte production level of LPS [13,14]. The above conclusions were based on the comparison of pregnancy outcomes between the DouStim protocol and one single COS cycle. Our study demonstrated that, compared with the two continuous mild stimulations protocol, the DouStim protocol could not significantly increase the production of oocytes and embryos even if the duration and dose of Gn were amplified in POR patients.

In this study, in the case of similar basic indicators such as the number of frozen embryos transferred and the number of FETs, there was no difference between the two groups in terms of pregnancy outcomes, including CPR and CLBR, as well as pregnancy complications such as early abortion rate, ectopic pregnancy rate, premature birth rate, and birth defect rate. Recently, Lu’s meta-analysis summarized recent studies on LPS, including 4433 patients. The results showed that there were similar results in pregnancy outcomes, indicating that LPS was not inferior to traditional follicular stimulation [15]. Yet the evidence for the wide application of LPS is insufficient. Cimadomo and Ubaldi et al. studied LPS and FPS at the genetic level and concluded that there was no significant difference in the gene expression of oocyte quality and in the follicular microenvironment [16,17]. Other experimental studies have shown that LPS-stimulated cumulus cells differ from traditional ovarian stimulation in gene expression (inflammation, oxidative phosphorylation, and apoptosis), which may have a negative impact on the mitochondrial function and immune response of oocytes and increase oocyte apoptosis and abnormal glucose metabolism. However, there was no significant difference between them in clinical observation indexes and pregnancy outcomes [18]. Chen et al. conducted a retrospective study on LPS-induced pregnancy and delivery and found no significant difference in congenital malformation rate, neonatal defect rate, and neonatal weight [19]. These studies and our study have confirmed that LPS will not damage pregnancy outcomes. The birth defect rate was 12% in the DouStim group and 5.9% in the two consecutive mild stimulations group, but there was no statistical difference (*p* > 0.05). We considered that the high value of the DouStim group was due to the small number of live birth samples. Therefore, if there are two or more antral follicles with a diameter of less than 8mm in the bilateral ovaries of POR patients on the trigger day of FPS, biphasic stimulation is still a feasible option. 

Depending on univariate analysis or clinical practice, several variables were included in the multivariate logistic regression for further analysis. This study found that age was a risk factor for pregnancy outcomes, while different COS protocols and other baseline data had no impact. This indicates that no matter what stimulation protocol is used, the pregnancy outcomes of elderly patients with POR will be worse. Further subgroup comparison based on age showed that there was no difference in clinical observation indicators and pregnancy outcomes between the two stimulation protocols in the young POR group. Although there was no significant difference in the yield of oocytes [20], the number of frozen embryos in the DouStim group was significantly lower than that in the two consecutive mild stimulations group in the elderly POR group (ages ≥ 35 years). This result indicates that the differences between the two stimulation protocols mainly occurred in the elderly subgroup.

There are some limitations in this study. As a retrospective study with a relatively small sample size, this study inevitably has selection bias. Furthermore, due to the limitation of the observation time window and patient selection, the pregnancy outcomes after all embryo transfers could not be observed.

## 5. Conclusions

This study shows that the DouStim protocol is inferior to the two consecutive mild stimulations protocol in terms of the number of frozen embryos, and this difference mainly occurs in elderly patients, but there is no difference in pregnancy outcomes between the two protocols. Therefore, DouStim remains an option worth considering under certain circumstances, such as when the ovarian response in FPS is poor and there is a considerable quantity of antral follicles on the trigger day, and in oncological patients who need fertility preservation urgently. In the future, a prospective randomized controlled trial (RCT) with multiple centers and large samples is needed to further verify the conclusions.

## Figures and Tables

**Table 1 jcm-11-06780-t001:** Comparison of basic characteristics between the DouStim group and the two consecutive mild stimulations group.

Variable	DouStim	Two Consecutive Mild Stimulations	
	N = 89	N = 192	*p*
Age (y)	33.87 ± 4.51	34.40 ± 4.03	0.323
BMI (kg/m^2^)	21.96 ± 2.87	22.39 ± 3.22	0.281
Infertility duration (y)	2 (3, 5)	3 (2, 5)	0.721
Infertility type (primary/secondary)	57/32	107/85	0.188
AMH (ng/mL)	0.60 (0.24, 0.89)	0.43 (0.20, 0.70)	0.081
Basal FSH (IU/L)	9.11 (7.01, 11.23)	9.85 (7.44, 13.15)	0.094
Basal LH (IU/L)	4.33 (3.30, 5.62)	4.50 (3.11, 6.18)	0.500
Basal E_2_ (pg/mL)	43.00 (29.31, 67.77)	9.85 (7.44, 13.15)	0.205
AFC (n)	5.64 ± 2.22	4.50 (3.11, 6.18)	0.315
Fertilization type, (IVF/ICSI)	62/27	123/69	0.357

Variables conforming to normal distribution are expressed as mean ± SD, and Student’s *t*-test was used for comparison between groups. Non-normally distributed continuous variables are expressed as median (25th to 75th quantiles) [M (P25, p75)], and comparisons were performed using the Mann–Whitney U test. The chi-square test was used for categorical variables. The basal FSH, LH, E_2_, and AFC values of the two groups were derived from the first ovarian stimulation cycle in the follicular phase.

**Table 2 jcm-11-06780-t002:** Comparison of clinical observation indexes between the DouStim group and the two consecutive mild stimulations group.

Variable	DouStim	Two Consecutive Mild Stimulations	
	N = 89	N = 192	*p*
Gn duration (d)	13.19 ± 4.41	11.03 ± 4.93	<0.01
Gn dosage (IU)	2250 (1800, 2850)	825 (525, 1200)	<0.01
Number of follicles (>14 mm)	2 (1, 3)	2 (1, 3)	<0.01
LH level on trigger day (IU/L)	5.87 (3.85, 9.49)	8.79 (6.02, 12.19)	<0.01
E2 level on trigger day (pg/mL)	753.0 (471.5, 1216.3)	561.5 (358.0, 845.0)	<0.01
P4 level on trigger day (ng/mL)	0.75 (0.27, 18.08)	0.27 (0.18, 0.39)	<0.01
Oocytes retrieved (n)	3.01 ± 2.187	3.42 ± 2.212	0.147
Number of 2PNs (n)	2 (0.5, 3)	2 (1, 3)	0.147
Number of frozen embryos (n)	1 (0, 2)	1 (0, 2)	0.035
Proportion of patients without frozen embryos (%)	39.3 (35/89)	26.0 (50/192)	0.024

**Table 3 jcm-11-06780-t003:** Comparison of pregnancy outcomes between the DouStim group and the two consecutive mild stimulations group.

Variable	DouStim	Two Consecutive Mild Stimulations	
	N = 89	N = 192	*p*
Times of FETs	1 (0, 1)	1 (0, 1)	0.561
Cumulative number of transferred embryos in two COS cycles (n)	1 (0, 2)	1 (0, 1)	0.512
Proportion of NCs (%)	11.8 (9/76)	6.9 (12/175)	0.190
Proportion of blastocysts transferred (%)	26.5 (22/83)	26.9 (53/197)	0.945
Proportion of patients with residual embryos to be transferred (%)	10.1 (9/89)	18.2 (35/192)	0.082
CPR (%)	31.5 (28/89)	36.5 (70/192)	0.413
CLBR (%)	28.1 (25/89)	26.6 (51/192)	0.789
Early abortion rate (%)	10.7 (3/28)	24.3 (17/70)	0.132
Ectopic pregnancy rate (%)	0.0 (0/26)	2.9 (2/70)	0.366
Neonatal preterm birth rate (%)	20.0 (5/25)	13.7 (7/51)	0.481
Birth defect rate (%)	12.0 (3/25)	5.9 (3/51)	0.353

One patient in the DouStim group experienced intrauterine fetal death at 32 weeks of gestation after FET, so one patient in the DouStim group who finally obtained live birth is missing. Birth defects included congenital heart disease and hypospadias.

**Table 4 jcm-11-06780-t004:** Multivariate logistic regression analysis of clinical pregnancy rate.

	B	Wald	*p*	OR	95% CI
Age	−0.101	9.052	0.003	0.904	0.847–0.966
AMH	0.550	1.799	0.180	1.733	0.776–3.873
bFSH	0.000	0.000	0.996	1.000	0.959–1.043
Infertility duration	0.043	0.890	0.345	1.044	0.955–1.142
Infertility types	−0.026	0.009	0.926	0.974	0.566–1.678
Stimulation protocol	−0.339	1.422	0.233	0.712	0.408–1.244

**Table 5 jcm-11-06780-t005:** Multivariate logistic regression analysis of live birth rate.

	B	Wald	*p*	OR	95% CI
Age	−0.099	7.710	0.005	0.905	0.844–0.971
AMH	0.844	3.628	0.057	2.327	0.976–5.548
bFSH	0.016	0.513	0.474	1.016	0.972–1.063
Infertility duration	0.020	0.157	0.692	1.020	0.924–1.126
Infertility types	−0.347	1.299	0.254	0.707	0.389–1.284
Protocols	−0.067	0.05	0.824	0.935	0.518–1.687

**Table 6 jcm-11-06780-t006:** Comparison of clinical observation indexes and pregnancy outcomes between the DouStim group and the two consecutive mild stimulations group in young POR patients (ages < 35 years).

Variable	DouStim	Two Consecutive Mild Stimulations	
	N = 44	N = 92	*p*
Oocytes retrieved (n)	3.18 ± 2.49	3.37 ± 2.19	0.656
Number of 2PNs (n)	2 (0, 3)	2 (1, 4)	0.375
Number of frozen embryos (n)	1 (0, 3)	1 (0, 2)	0.820
Number of embryos transferred (n)	1 (0, 2)	1 (0, 2)	0.402
Times of embryos transferred (n)	1 (0, 1.75)	1 (0, 1)	0.463
CPR (%)	40.9 (18/44)	44.6 (41/92)	0.687
LBR (%)	38.6 (17/44)	33.7 (31/92)	0.573

**Table 7 jcm-11-06780-t007:** Comparison of clinical observation indexes and pregnancy outcomes between the DouStim group and the two consecutive mild stimulations group in elderly POR patients (ages ≥ 35 years).

Variable	DouStim	Two Consecutive Mild Stimulations	
	N = 45	N = 100	*p*
Oocytes retrieved (n)	2.84 ± 1.86	3.47 ± 2.24	0.104
Number of 2PNs (n)	2 (1, 3)	2 (1, 3)	0.243
Number of frozen embryos (n)	1 (0, 1.5)	1 (0.25, 2)	0.005
Number of embryos transferred (n)	1 (0, 1)	1 (0, 2)	0.068
Times of embryos transferred (n)	1 (0, 1)	1 (0, 1)	0.108
CPR (%)	22.2 (10/45)	29.0 (29/100)	0.394
LBR (%)	17.8 (8/45)	20.0 (20/100)	0.755

## Data Availability

The data presented in this study are available upon request from the corresponding author. The data are not publicly available due to privacy concerns related to protected health information.

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
