# Peer review of "Pregnancy Outcomes in Double Stimulation versus Two Consecutive Mild Stimulations for IVF in Poor Ovarian Responders"

_jcm, 2022, doi:10.3390/jcm11226780_

Round 1

Reviewer 1 Report

In this manuscript, authors are describing and compare two approaches in ovarian stimulation in poor responders. The results are interesting and in the interest of the readers, but there are some issues that need to be clarified. 

Comments:

- line 78: I don't think that teratozoospemia (deformation rate over 98%) should be excluded. The latest data suggest that morphology isn't so important factor as it was suggested before. Especially using ICSI this hurdle is overcomeline. 

- line 92: you wrote that ovarian stimulation in LP was started on the second day after oocyte retrieval. How is that? Usually, it is started on day 5 after oocyte retrieval. This might affect an inferior outcome compared to two consecutive mild stimulation. Please discuss

- lines 110-112: it is understandable that ivf centers have different approaches when to transfer embryos but when analyzing data, this should be considered. Therefore I suggest you provide data on the outcome when embryos were transferred at cleavage or at blastocys stage. 

- line 150: ....used as the molecule...? I suggest to rewrite this sentence

-table 2: P values cannot be presented as 0.000

-table 3: define birth defects. It seems rather high percentage. Compare to other studies in discussion. 

-table 7: define how CPR and LBR were calculated. Per transfer? Per patient/cycle?

- you suggest that duostim is inferior to two consecutive mild protocol, but there still might be cases when such approach might be highly suggested. For instance in oncological patiens when there is no time to conduct two consecutive stimulations. What is you opinion about this? Please write about this in discussion.

Reviewer 2 Report

Thank you for allowing me to review this article comparing duo stim to two consecutive stimulation cycles.

Introduction

In the introduction the authors state: " . Thus, the mild-stimulation protocol with low-dose Gn and estrogen antagonist (like clomiphene) as adjuvants remain the most cost-effective and recommended approach to POR patients"

While the cited guideline does state that mild stimulation is cost effective (based on one study) it does not recommend any specific protocol for treatment and even states: In women considered to be poor responders, there is insufficient evidence to recommend for or against IVF with mild ovarian stimulation using oral agents alone over conventional gonadotropin stimulation. I suggest that the sentence should be rephrased and not state that  mild stimulation protocol with estrogen antagonist is the recommended protocol.

The aim of this study is nicely written in the introduction

Materials and methods

"with POR undergoing oocyte retrieval for IVF or intracytoplasmic sperm injection (ICSI) according to Bolognia criteria were recruited"

Recommend rephrasing to: with POR according to Bologna criteria undergoing oocyte retrieval for IVF or intracytoplasmic sperm injection 

Please note Bologna and not Bolognia

While it is reasonable could the authors explain why they chose the Bologna criteria and not Poseidon?

Line 72 - please correct typo - stimulation and not stimution

Is there a reason for collection 34 hours after trigger in the first stimulation cycle and 36 hours after trigger in the second duostim cycle?

107-109 -please explain the grading system of the blastocysts (was it grading according to gardner schoolcraft)?

Is it the clinics protocol to not attempt fresh transfers? If not and sometimes fresh transfers are attempted this may cause a selection bias in cases of 2 consecutive stimulation cycles) 

Was it the clinics protocol to transfer Cleavage-stage embryos and blastocysts  at the third or fifth day after hCG - as opposed to after ovulation in the natural cycle?

Results-

Basal LH and basal FSH are missing in table 1 for the two consecutive stimulation patients

Do the results in table 2 refer to the first stimulation cycle? It is not clear as some results cannot be for two cycles...

Results pertaining to table 3, table 6 and 7 - were power calculation done/

There is no section of limitations in the discussion section. Please add this to the discussion. 

Some grammatical issues, please correct

This article aims to answer an important question - is duostim better than 2 consecutive stimulation cycles. While this is an important question that seems to be answered by the authors there are revisions required before this article is published

Round 2

Reviewer 1 Report

have no further review